# Genetic Biomarkers of Antipsychotic-Induced Prolongation of the QT Interval in Patients with Schizophrenia

**DOI:** 10.3390/ijms232415786

**Published:** 2022-12-13

**Authors:** Elena E. Vaiman, Natalia A. Shnayder, Nikita M. Zhuravlev, Marina M. Petrova, Azat R. Asadullin, Mustafa Al-Zamil, Natalia P. Garganeeva, German A. Shipulin, Paul Cumming, Regina F. Nasyrova

**Affiliations:** 1Institute of Personalized Psychiatry and Neurology, V. M. Bekhterev National Medical Research Centre for Psychiatry and Neurology, 192019 Saint Petersburg, Russia; 2Shared Core Facilities “Molecular and Cell Technologies”, V. F. Voyno-Yasenetsky Krasnoyarsk State Medical University, 660022 Krasnoyarsk, Russia; 3Department of Psychiatry and Addiction, Bashkir State Medical University, 450008 Ufa, Russia; 4Department of Physiotherapy, Faculty of Continuing Medical Education, Peoples’ Friendship University of Russia, 117198 Moscow, Russia; 5Department of General Medical Practice and Outpatient Therapy, Siberian State Medical University, 634050 Tomsk, Russia; 6Centre for Strategic Planning and Management of Biomedical Health Risks Management, 119121 Moscow, Russia; 7Department of Nuclear Medicine, Bern University Hospital, 3010 Bern, Switzerland; 8School of Psychology and Counselling, Queensland University of Technology, Brisbane 4000, Australia; 9International Centre for Education and Research in Neuropsychiatry, Samara State Medical University, 443016 Samara, Russia

**Keywords:** schizophrenia, antipsychotics, adverse drug reaction, antipsychotic-induced repolarization disorder, cardiac repolarization, long QT interval, sudden death syndrome, ventricular tachyarrhythmia, pharmacogenetics

## Abstract

Antipsychotics (AP) induced prolongation of the QT interval in patients with schizophrenia (Sch) is an actual interdisciplinary problem as it increases the risk of sudden death syndrome. Long QT syndrome (LQTS) as a cardiac adverse drug reaction is a multifactorial symptomatic disorder, the development of which is influenced by modifying factors (APs’ dose, duration of APs therapy, APs polytherapy, and monotherapy, etc.) and non-modifying factors (genetic predisposition, gender, age, etc.). The genetic predisposition to AP-induced LQTS may be due to several causes, including causal mutations in the genes responsible for monoheme forms of LQTS, single nucleotide variants (SNVs) of the candidate genes encoding voltage-dependent ion channels expressed both in the brain and in the heart, and SNVs of candidate genes encoding key enzymes of APs metabolism. This narrative review summarizes the results of genetic studies on AP-induced LQTS and proposes a new personalized approach to assessing the risk of its development (low, moderate, high). We recommend implementation in protocols of primary diagnosis of AP-induced LQTS and medication dispensary additional observations of the risk category of patients receiving APs, deoxyribonucleic acid profiling, regular electrocardiogram monitoring, and regular therapeutic drug monitoring of the blood APs levels.

## 1. Introduction

Patients with Sch have higher rates of morbidity and mortality, compared to the general population, including an especially elevated risk of death from cardiovascular diseases (CVDs) [1]. Patients with Sch require long-term use of antipsychotics (APs) in mono- or polytherapy, which leads to the development of adverse reactions (ADRs), including metabolic syndrome [2] and cardiac disturbances. Such reactions are a secondary cause of increased mortality in Sch [2]. As described with other psychotropic drugs, such as antidepressants, APs can cause the prolongation of the QT interval [3]. In the 1960s, electrocardiogram (ECG) investigations first documented long QT syndrome (LQTS) in patients with Sch who were under treatment with APs [4,5,6,7].

The QT interval reflects the depolarization and repolarization cycle of the heart ventricles, which is signaled by the action potential in cardiomyocytes [8]. Normally, the duration of the QT interval is less than 450 ms in women and less than 430 ms in men [9]. QT interval lengthening leads to such ADRs as impairment of cardiac conduction [10,11], atrial fibrillation [12], arterial hypotension/hypertension [13], myocarditis [14], malignant ventricular tachyarrhythmia (Torsade de pointes—TdP) [15], and sudden death syndrome (SDS) [16]. In patients receiving APs, the incidence of SDS or ventricular arrhythmias is in the range of 1.7 to 5.3% [17,18].

In general, psychiatric patients under medication show only minor QT interval prolongation. In a study by Novotny et al. [19], only 2% of patients with Sch had LQT while taking APs for 2 years. Nevertheless, the management of ADRs remains a serious problem in the clinical practice for attending psychiatrists and cardiologists. LQTS can be familial or acquired, as in the case of susceptible patients under treatment with APs. Familial LQTS includes Jervell and Lange-Nielsen syndrome (JLNS) (autosomal recessive form), which is characterized by congenital sensorineural hearing loss and a high incidence of SDS in childhood, as well as Romano–Ward syndrome (autosomal dominant form), which is not associated with sensorineural hearing loss [20].

Mutations in the *KCNQ1* (potassium voltage-gated channel subfamily Q member 1) gene and *KCNE1* (potential-dependent potassium channel H2 type) gene encoding subunits of certain voltage-gated potassium channels are associated with the JLNS phenotype. These two genes code for proteins that mediate potassium flux (i.e., depolarization) in cells of the heart and inner ear [5]. The prevalence of JLNS among populations of children with congenital sensorineural is in the range of 0 to 2.6% [5,6,7,8,9]. Susceptibility to acquired LQTS is also genetically determined [21], with a heritability of approximately 35% in the general population (excluding those patients with congenital LQTS) [22,23]. First-degree relatives of patients with congenital LQTS have a higher risk of AP-induced LQTS [24]. Thus, the management of LQTS is an interdisciplinary problem arising in clinical psychiatric practice.

The aim of this narrative review is to formulate a generalization of the results of studies investigating genetic biomarkers associated with AP-induced LQTS in patients with Sch.

## 2. Genetic Predictors of Long QT Syndrome

### 2.1. Causal Genes for Familial Long QT Syndrome

Although the diagnosis and management of patients with familial forms of LQTS are the prerogative of cardiologists, attending psychiatrists must be mindful of the possibility that their patients may be carriers of the gene mutations responsible for familial forms of LQTS, thus increasing their risk for iatrogenic cardiac events. Despite the low comorbidity of Sch and familial forms of LQTS, the possibility must be considered at the onset of AP treatment. However, there are presently no algorithms for an interdisciplinary approach to managing such an unfavorable SDS phenotype in a psychiatric interdisciplinary practice. At the same time, with the improvement of molecular genetic diagnostics, panels for deoxyribonucleic acid (DNA) profiling of patients with SDS or a burdened family history for LQTS have been developed and introduced into clinical practice. Therefore, we recommend that the family history and anamnesis of cardiac health is necessary before the onset of long-term psychiatric treatment with APs. The most common forms of LQTS are shown in Table 1.

The diversity of genetic factors involved in LQTS is of considerable practical importance in psychiatric practice, since the development of monogenic forms of LQTS can occur independently of external environmental factors, such as the use of APs. In this regard, the risk of SDS is elevated even in schizophrenic patients receiving monotherapy with APs that affect heart rate and conduction (Table 2), which calls for careful consideration of dose and particular APs, along with the avoidance of high-risk drugs, such as haloperidol or chlorpromazine (Figure 1).

### 2.2. Candidate Genes Responsible for Prolongation of the QT Interval

In the general population and among patients with Sch, the prevalence of multifactorial and polygenic forms of LQTS is higher than that of monogenic forms. Such multifactorial forms are associated with the carriage of single nucleotide variants (SNVs) of various candidate genes predisposing to the prolongation of the QT interval, subject to additional environmental factors, such as treatment with APs. There are increasing numbers of associative genetic studies of SNVs of candidate genes encoding ion channels expressed both in the central nervous system (CNS) and in the heart. This progress makes the identification of patients with Sch who are at risk for SDS with long-term use of APs more complicated (Table 3).

#### 2.2.1. The *KCNH2* (*HERG*) Gene

The *KCNH2* gene is located on chromosomes 7q35–36 and encodes the fast internal rectifying potassium ion channel (IKr). This gene expression is predominantly in endocrine tissues and bone marrow but is also present in the heart muscle (Figure 2). Mutations in the *KCNH2* gene decrease the current of potassium ions per channel, and thus lengthen the action potential in cardiomyocytes and neurons [51]. According to Khera et al. [47], SNV rs189014161 (3591 C > T) of the *KCNH2* gene is associated with increased risk of LQTS prolongation.

#### 2.2.2. The *SCN5A* Gene

The *SCN5A* (alpha subunit of voltage-gated type 5 sodium channel) gene is located on chromosome 3p22.2 and encodes the alpha subunit of the cardiac voltage-gated type 5 sodium channel. This gene is expressed predominantly in skeletal muscles and in myocardium (Figure 3). The activation of this ion channel leads to a rapid depolarization and increased amplitude of the cardiac action potential. Some SNVs of the *SCN5A* gene increase the risk of developing SDS [54].

Spellmann et al. [48] observed a significant effect of the major allele of SNV rs1805124 (50744 A > G) of the *SCN5A* gene on QTc prolongation. Their data suggested a common association between the genotypes and QT interval length. So, homozygous AA genotype was associated with a shorter QT interval. Gouas et al. [57] found an association between allelic variants of SNV rs1805124 and QT prolongation in healthy individuals, where carriers of the G allele showed a longer QT interval. On the other hand, Hobday et al. [58] found no such association in a cohort of patients with CVD. Similarly, Lehtinen et al. [59] failed to find a significant effect of SNV rs1805124 of the *SCN5A* gene on the duration of the QT interval in patients with diabetes mellitus. Although Pfeufer et al. [60] did not find a significance effect of this SNV in a genome-wide analysis, they did see evidence for a small independent effect on QTc duration.

#### 2.2.3. The *KCNE1* (*MiRP1*) Gene

The *KCNE1* (potential-dependent potassium channel E1 type) gene (also known as *MIRP1)* is located on chromosome 21q22 and encodes a voltage-gated slow-type potassium channel (IKs). These channels have their predominant expression in the brain, ovary/uterus, and muscles, including the myocardium (Figure 4). Mutations in this gene can alter the channel opening and closing kinetics, thereby reducing potassium currents into the cell. Some SNVs in the *KCNE1* gene predispose to the development of AP-induced LQTS [61].

Alternative microvoltage T-waves on the ECG are characterized by fluctuations in T-wave amplitude and morphology between heartbeats. This is an electrophysiological phenomenon that is clinically associated with ventricular arrhythmias and an important marker of arrhythmia risk. Koskela et al. [64] hypothesized that the same SNVs affecting the QT interval also affect microvoltage T-wave variation. They studied the association of the rs1805127 (66793 A > G, 6793A > C) and rs727957 (8542 C > T) of the *KCNE1* gene and rs1805124 (50744 A > G) of the *SCN5A* gene with microvoltage T-wave variability during a clinical exercise test. Their main results were that microvoltage T-waves were lowest at all stages of the exercise test in patients carrying the homozygous TT genotype (rs1805127) of the *KCNE1* gene. This finding remained statistically significant after adjusting for age, history of coronary artery disease, and chronic use of beta-blockers [64].

#### 2.2.4. The *AKAP9* (*KCNQ1*) Gene

The *AKAP9* (A-Kinase Anchoring Protein 9) gene (also known as *KCNQ1)* is located on chromosome 7q21.2 and encodes an alpha-kinase type 9 anchor protein. It has varying degrees of expression in all organs (Figure 5). AKAP9 forms a macromolecular complex with the α-subunits of the voltage-gated potassium channel, Kv7.1 (also known as KCNQ1) and its associated β-subunits (KCNE1), which are part of the IK complex [65]. AKAP binds directly with KCNQ1 and promotes its phosphorylation [66], and it is itself a substrate for phosphorylation, all of which influences the activity of voltage-gated potassium channels.

de Villiers et al. [50] investigated the association of four SNVs (rs11772585, rs7808587, rs2282972, and rs2961024) of the *AKAP9* gene with the risk of QT prolongation. They found that the carriage of the homozygous GG genotype (rs2961024, 125,882 G > C), which is more often represented by homozygotes for the CGCG haplotype, led to an age-dependent prolongation of the QT interval (increasing by 1% per decade) (*p*-value = 0.006). The T-allele (rs11772585, 16,315 C > T), uniquely found in the TACT haplotype, more than doubled (218%) the risk of CVD (*p*-value = 0.002) in carriers of the SNV rs12720459 (1022 C > T). In addition, this aggravated the CVD severity (*p*-value = 0.025). The GG genotype (rs7808587, 54,908 G > A) was associated with 74% increased risk of cardiac arrhythmias (*p*-value = 0.046), while the carriage of the T allele (rs2282972, 1181 C > T), predominantly represented by the CATT haplotype, reduced the risk of LQTS by 53% (*p*-value = 0.001).

Thus, the DNA profiling of patients with Sch to identify the SNVs of candidate genes predisposing to multifactorial forms of LQTS can help identify patients at risk. Such individuals would require ECG monitoring and consultation with a cardiologist (according to indications) while taking AP in monotherapy, and especially so in polytherapy (Figure 6).

### 2.3. Candidate Genes Responsible for Poor Metabolism of Antipsychotics

Most APs of first and second generations are metabolized by hepatic cytochrome P450 isoenzymes [69,70]. Four pharmacogenetic phenotypes are distinguished depending on the rate of drug metabolism: extensive (EM), intermediate (IM), poor (PM), and ultrarapid (UM) metabolizers [71]. From the standpoint of the LQTS problem, individuals of the PM phenotype call for particular attention, since such patients can inadvertently experience subtoxic or toxic levels of AP in the blood serum. In that circumstance, they are at particular risk of developing secondary LQTS against the background of acute or chronic AP intoxication, even if they are not carriers of gene mutations responsible for monogenic forms of LQTS or SNVs of candidate genes predisposing to multifactorial forms of LQTS. The most significant cytochrome P450 isoenzymes involved in APs metabolism are CYP2D6, CYP3A5, and CYP1A2 (Table 4).

#### 2.3.1. The *CYP2D6* Gene

The *CYP2D6* (cytochrome P450 family 2 subfamily D member 6) gene is located on chromosome 22q13.1. Its highest expression is in the liver and gastrointestinal tract (Figure 7). This gene is the most studied gene of the cytochrome P450 family in relation to the prolongation of the QT interval induced by APs [75].

Thus, the metabolism of thioridazine to mesoridazine is mediated by the CYP2D6 isoenzyme, while the further metabolism of mesoridazine to sulforidazine is mediated by another isoenzyme [78]. An individual’s activity of the CYP2D6 isoenzyme determines the level of thioridazine and its metabolites in blood plasma [78,79] and may affect the risk of developing cardiovascular ADRs, including LQTS. Nearly 5–10% of Europeans are of the PM phenotype [80]. Therefore, it is advisable to conduct therapeutic drug monitoring in patients with Sch taking APs that have predominant metabolism via the CYP2D6 isoenzyme; these APs include risperidone, quetiapine, olanzapine, aripiprazole, thioridazine, perphenazine, haloperidol, and chlorpromazine.

In a case reported by Mazer-Amirshahi. et al. [81], a 14-year-old boy intentionally swallowed 20 × 5 mg aripiprazole tablets. Initially, the ECG showed sinus tachycardia with a QRS interval duration of 138 ms and a QT interval of 444 ms. The results of genetic testing performed during hospitalization showed that the patient was of the PM phenotype, i.e., a homozygous carrier of a non-functional variant *4 of the *CYP2D6* gene. This case report substantiates the threat that aripiprazole toxicity may provoke QT interval prolongation, albeit without associated arrhythmias or a personal or family history of cardiovascular events.

In general, the cardiotoxicity of APs may increase in patients with the PM phenotype for CYP2D6 [81]. This holds for carriers of non-functional SNVs of the *CYP2D6* gene: *2XN, *3A, *3B, *4, *4F, *4G, *4H, *4xN, *5, *6, *7, *8, *11, *12, *14, *15, *18, *19, *20, *21, *38, *40, *42, *44, and *56 [71]. The most common non-functional SNV of the *CYP2D6* gene the in the European population is *4.

#### 2.3.2. The *CYP3A5* Gene

The *CYP3A5* (cytochrome P450 family 3 subfamily A member 5) gene is located on chromosome 7q22.1 and encodes the 3A5 isoenzyme of the liver. It is expressed most in the liver and gastrointestinal tract (Figure 8).

Risperidone is metabolized by the CYP3A5 isoenzyme encoded by the gene of the same name. According to Ranjbar et al. [73], risperidone treatment had a significant effect on the QT interval and QT variance (*p*-value < 0.01). Schizophrenic patients with the CYP3A5 PM phenotype who take haloperidol, risperidone, and aripiprazole have an increased risk of AP-induced LQTS. Risk assessment calls for DNA profiling of carriage of various non-functional alleles of the *CYP3A5* gene: *2, *3, *6, *7, *8, *9, *10, *11, *3D, *3F, 3705 C > T, and 7298 C > A. The most common non-functional SNV of the *CYP2D6* gene in the European population is *3.

#### 2.3.3. The *CYP1A2* Gene

The *CYP1A2* (cytochrome P450 family 1 subfamily A member 2) gene is located on chromosome 15q24.1 and encodes the liver isoenzyme 1A2. It is exclusively expressed in the liver (Figure 9).

Olanzapine is metabolized by the CYP1A2 isoenzyme encoded by the gene of the same name. According to Shoja Shafti et al. [74], risperidone treatment had a significant effect on the QT interval (*p*-value < 0.01). Schizophrenic patients with the CYP3A5 PM phenotype who take olanzapine, clozapine, thioridazine, and chlorpromazine have an increased risk of AP-induced LQTS. Risk assessment calls for DNA profiling of the carriage of non-functional alleles of the *CYP1A2* genes, including SNVs: *1C, *1K (−729 C > T or −739 T > G), *3, *4, *6, and *7 (3534 G > A).

From a practical perspective, patients with SNVs of candidate genes that predispose to the reduced metabolism of APs, and thus increased risk of acute/chronic intoxication, including cardiotoxicity and secondary LQTS, require dynamic TDM of their prescribed APs, especially in the monotherapy mode (Figure 10). At the same time, the prescription of APs with a similar (competing) metabolic pathway in patients with SDSs who have a PM profile for the above non-functional SNVs should be avoided.

## 3. Discussion

The problem of drug-induced LQTS was a discussion topic at the 2005 International Conference on Harmonization of Technical Requirements for Registration of Pharmaceuticals for Human Use (ICH). This led to the adoption and publication of recommendations on the evaluation of elongation QT interval and proarrhythmic potential of non-antiarrhythmic drugs (ICH-E14) [86,87]. The study concluded that almost all APs, with a few exceptions, should undergo a “thorough QT study” (TQT), as defined by ICH-E14. However, most of the APs presently in use (predominantly of the first generation) entered the market long before TQT studies became mandatory. On the other hand, even a well-conducted TQT study cannot exclude all the risk of developing AP-induced LQTS if the AP is administered at high doses for an extended period, or in clinical situations of polypharmacy, substance abuse, or preexisting cardiovascular disease. In many respects, the risk of cardiac complications from APs is genetically determined (Table 3 and Table 4). As such, psychiatrists and general practitioners should be in a position to assess and manage the potential risk of LQTS and other cardiac arrhythmias and conduction disorders caused by APs [88].

Fanoe et al. [88] proposed classifying drugs used in psychiatry into three categories according to the reported effects at therapeutic plasma levels on the QT interval, the induction of cardiac arrhythmias, and conduction disturbances. This study made the predominant distinction between drugs with no risk of cardiac arrhythmias and conduction disorders (Class A drugs) and drugs with some degree of risk (Class B drugs). The latter group was further subdivided into class B* drugs (Table 5). APs that are best avoided in patients with congenital QT interval prolongation, with a conditional risk of TdP, and those with a known risk of TdP were compiled into class B* [2,87,88]. LQTS is an important risk factor for the development of TdP, which, in turn, can lead to ventricular fibrillation and SDS [89]. The relationship between AP-induced LQTS and TdP frequency is not direct [87,90,91], since many factors and comorbid conditions can increase or decrease the risk of these iatrogenic effects [90,92,93,94]. The analysis of first and new generation APs currently used to treat Sch, and many years of experience in our own clinical practice, indicate that there is no completely safe medication, i.e., no APs can be properly assigned to category A [95].

Our review of genetic biomarkers of the risk of developing a prolonged QT interval while taking APs in patients with Sch, especially in chronic (long-term) therapy, indicates the need for an interdisciplinary approach involving a psychiatrist, cardiologist, and clinical pharmacologist to screen for familial and multifactorial forms of LQTS (Figure 1, Table 1). This is especially the case when a combination of the carriage of candidate genes in a given patient with SNVs predisposing to the prolongation of the QT interval (Figure 9, Table 3) and the slowing of the metabolism of APs (Figure 10, Table 4). When a number of APs are prescribed, first- and new-generation APs with a high risk of SDSs are contraindicated. On the other hand, it is possible to prescribe some APs cautiously to patients with Sch with a genetic predisposition to prolongation of the QT interval (carriers of SNVs candidate genes predisposing to the development of multifactorial forms of LQTS) or with a genetic predisposition to the poor metabolism of APs in the liver by P-oxidation (for example, the phenotype “intermediate metabolizer”). In such cases, it is necessary to choose an AP with a low risk of drug-induced slowing of the QT interval, use AP monotherapy, conduct ECG and TDM in dynamics against the background of antipsychotic therapy (Figure 11 and Figure 12).

Based on the above summary, we present an algorithm for differential diagnosis of the risk for LQTS in patients with Sch (Figure 13).

In the first instance, a patient admitted to inpatient treatment for Sch (or their caregivers) must provide an anamnesis of their life prior to the illness, aiming to determine whether the patient has a history of LQTS (step I). If the anamnesis is not burdened by positive history, there should be ECG monitoring every six months while taking APs. Patients having a causal LQTS should have an ECG and Holter ECG monitoring before initiation of AP treatment. If baseline results are normal, ECG monitoring should be repeated every six months during AP therapy. If QTc prolongation is noted, there is a call for the genetic testing of risk factors, including CYP isozymes. All patients with Sch who present causal LQTS should undergo genetic testing prior to the initiation of AP therapy (step II). Further, according to the results of genetic testing, SNVs/polymorphisms should be registered in casual genes for LQTS, candidate genes responsible for LQTS, and candidate genes responsible for the slow metabolism of APs (PM and IM phenotypes).

Patients with an absence of causal mutations of LQTS based on the results of genetic testing should have repeated ECG every six months on APs therapy. Patients with a family history of LQTS must avoid APs known to interact with culprit genes or allelic variants. Patients with SNVs/polymorphisms in candidate genes responsible for LQTS should have ECG monitoring as standard practice every six months while on AP therapy. Patients with SNVs/polymorphisms in culprit genes should have repeated ECG every month while under treatment. Also, practitioners should be prepared to change treatment to APs with the lowest risk of causing LQTS.

In the case of genetic testing for candidate genes responsible for the slow metabolism of APs, patients should be triaged into groups with low, moderate, and high risk. In the low-risk group, APs can be prescribed at the typical therapeutic dosage, in keeping with their low risk of LQTS. Standard practice should include repeated ECG every six months and therapeutic drug monitoring once every 12 months. Patients in the moderate risk group can be prescribed an AP at 25–50% of the regular therapeutic dosage, in conjunction with biannual Holter ECG and therapeutic drug monitoring. Patients in the category with high risk of developing LQTS should be excluded from treatment with APs that are metabolized by hepatic enzymes with deficient activity. These high-risk patients should be treated with an AP that is metabolized by intact pathways, with monthly ECG examinations and therapeutic drug monitoring at intervals of three months.

Despite decades of research and progress in modern diagnostic methods in psychiatry, cardiology, clinical pharmacology, and medical genetics, AP-induced cardiac arrhythmias and conduction disorders remain a pressing problem for clinicians and patients with Sch. Documenting the carriage of SNVs of candidate genes presented in this narrative review, and many others awaiting study, will bring us closer to the goal of personalized psychiatry [96].

The identification of genetic variants that predispose to LQTS in patients with Sch who are treated with APs may help minimize their risk of iatrogenic LQTS and SDS, while also reducing the likelihood of the withdrawal of the effective APs from the pharmaceutical market. Medications that have off-target effects on ion channels in the CNS will likely also affect ion channel activity in the heart. Thus, psychiatrists, in consultation with cardiologists and medical geneticists, should attend to AP-induced ADRs and their unwanted effects on the heart. Regular ECGs in patients with Sch under chronic AP therapy may help to guide the switching over to an AP, bringing lesser risk of SDS. We hope that a multidisciplinary approach to predicting and preventing arrhythmias and conduction disturbances in patients with Sch may improve the safety of antipsychotic therapy, without penalty in the primary outcome of relieving psychotic symptoms.

There remain distinct limitations to the effective research on genetic links between cardiac arrhythmias and conduction disturbances and APs use in patients with Sch. These limitations include incomplete (missing) autopsies, lack of adequate antemortem and post-mortem DNA analysis in the cases of sudden and inexplicable death of patients with Sch. Another limiting factor is the persistent lack of informed consent of patients with Sch or their legal representatives to participate in genetic research. Indeed, most genetic studies of LQTS in patients with Sch are single cases or small case series without deep phenotyping or DNA profiling of the patients’ pedigrees, which calls for ethics approval and complicated logistics. Attending psychiatrists should always be mindful of the presence of comorbid genetically determined heart disease accompanied by cardiac rhythm and conduction disturbances, which significantly increases the risk of AP-induced LQTS or SDS in patients with Sch. Of greatest interest for future research is the carriage of mutations and SNVs in genes encoding voltage-gated sodium and potassium channels, especially those channels that are expressed both in the CNS and in the heart. All the above factors highlight the importance of the joint management of patients with Sch by a team including a psychiatrist, cardiologist, and clinical pharmacologist. Finally, we recommend the implementation in protocols of primary diagnosis and medication dispensary additional observations of the risk category of patients receiving AP treatment, DNA profiling, regular ECG monitoring, and regular therapeutic drug monitoring (TDM) of blood AP levels.

## 4. Conclusions

LQTS is one of the most important, life-threatening complications of antipsychotic therapy, the development of which reduces the patient’s compliance with therapy. The development of genetic diagnostic panels that can be used even before the start of therapy will improve the quality of treatment and the patient’s life.

## Figures and Tables

**Figure 1 ijms-23-15786-f001:**
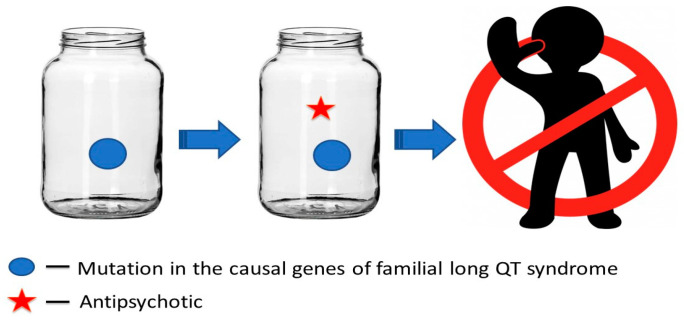
Monogenic forms of long QT syndrome: the use of antipsychotics that affect heart rate and conduction is contraindicated due to the high risk of sudden death syndrome, even in monotherapy.

**Figure 2 ijms-23-15786-f002:**
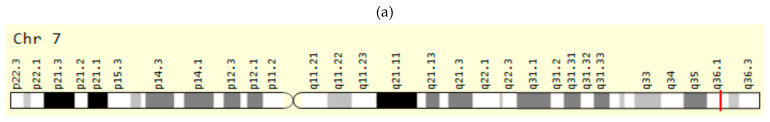
The *KCNH2 (HERG)* gene: (**a**)—gene chromosome location; (**b**)—protein expression in tissues of humans [52,53].

**Figure 3 ijms-23-15786-f003:**
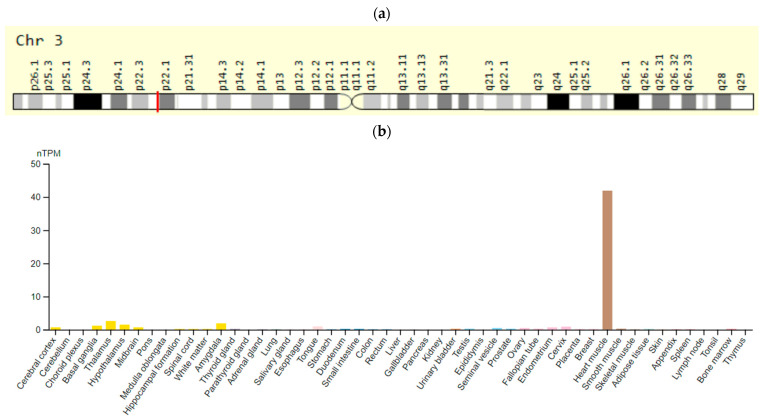
The *SCN5A* gene: (**a**)—gene chromosome location; (**b**)—protein expression in tissues of humans [55,56].

**Figure 4 ijms-23-15786-f004:**
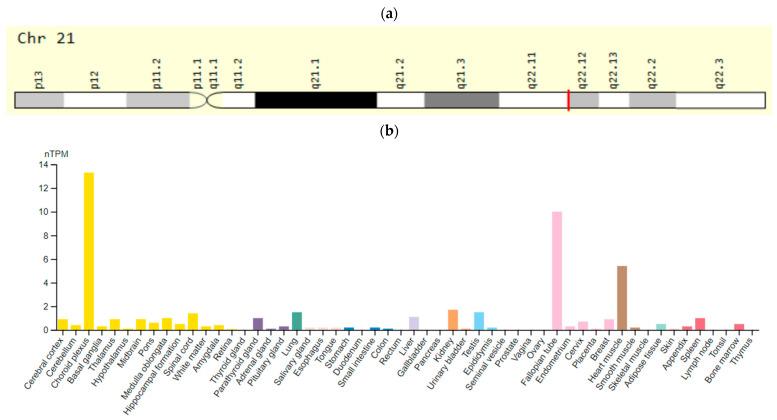
The *KCNE1 (MiRP1)* gene: (**a**)—gene chromosome location; (**b**)—protein expression in tissues of the human body [62,63].

**Figure 5 ijms-23-15786-f005:**
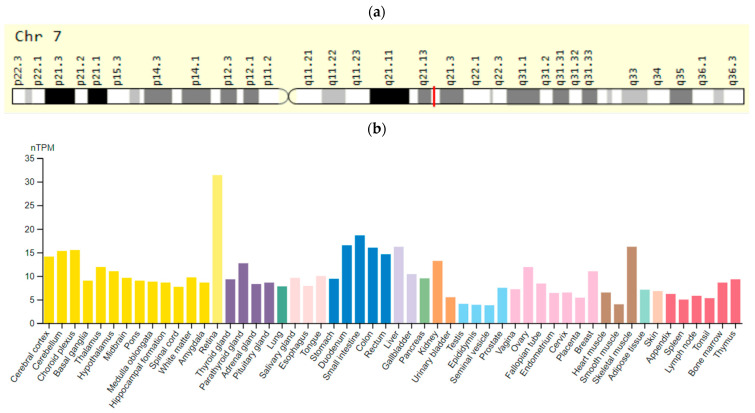
The *AKAP9 (KCNQ1)* gene: (**a**)—gene chromosome location; (**b**)—protein expression in tissues of the human body [67,68].

**Figure 6 ijms-23-15786-f006:**
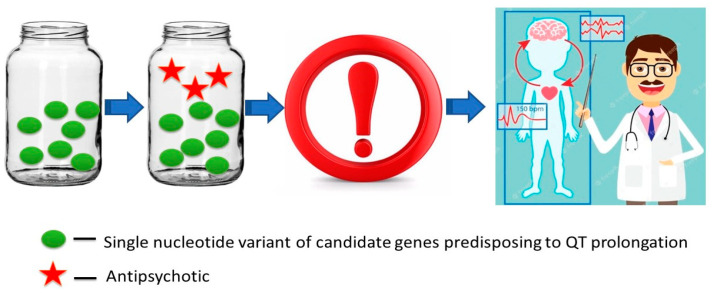
Multifactorial forms of long QT syndrome: the administrations of antipsychotic medications that affect heart rate and conduction calls for caution, especially in polytherapy. Here, dynamic Holter ECG monitoring is required.

**Figure 7 ijms-23-15786-f007:**
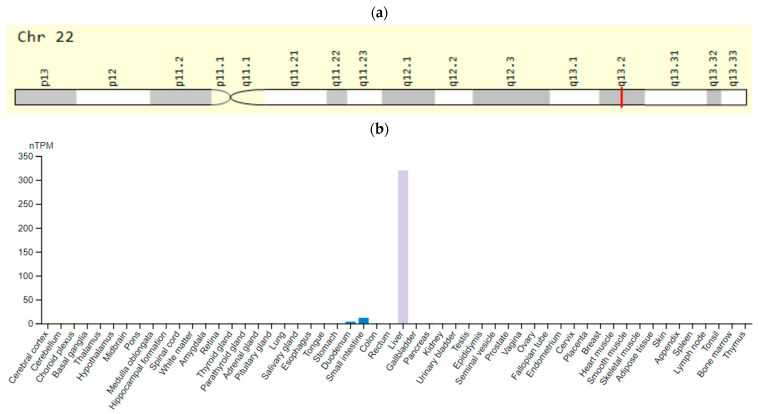
The *CYP2D6* gene: (**a**)—gene chromosome location; (**b**)—protein expression in tissues of the human body [76,77].

**Figure 8 ijms-23-15786-f008:**
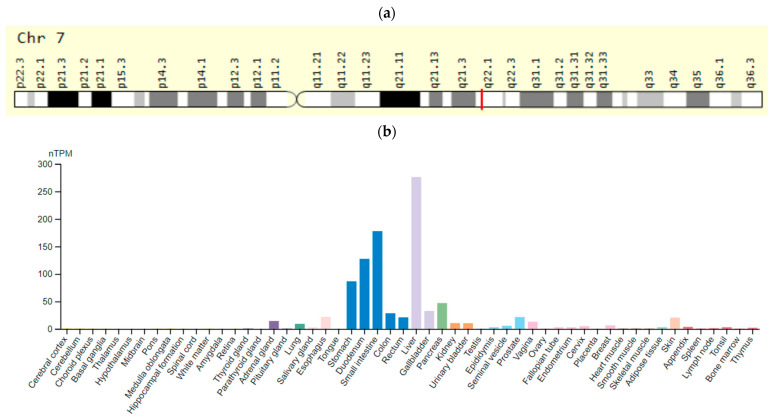
The *CYP3A5* gene: (**a**)—gene chromosome location; (**b**)—protein expression in tissues of the human body [82,83].

**Figure 9 ijms-23-15786-f009:**
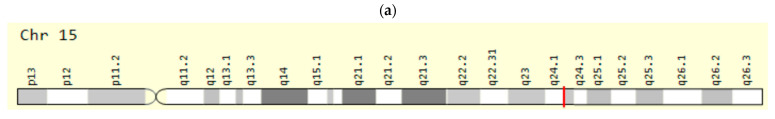
The *CYP1A2* gene: (**a**)—gene chromosome location; (**b**)—protein expression in tissues of the human body [84,85].

**Figure 10 ijms-23-15786-f010:**
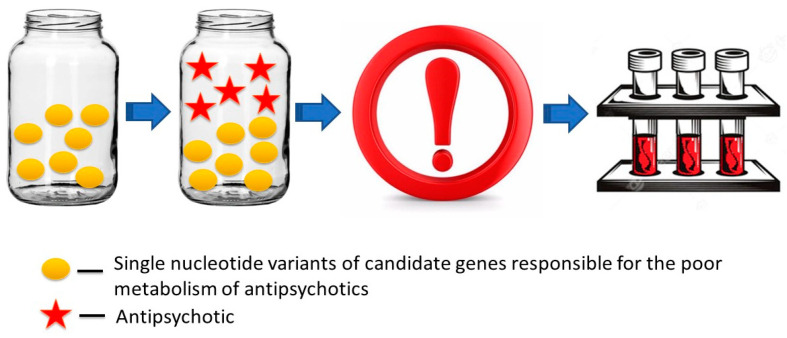
Poor metabolizer phenotype predisposing to high toxic blood levels of antipsychotics resulting in symptomatic QT interval prolongation. Antipsychotics that affect heart rate and conduction may be used, but with caution, and preferably in monotherapy, with dynamic therapeutic drug monitoring of the plasma antipsychotic levels.

**Figure 11 ijms-23-15786-f011:**
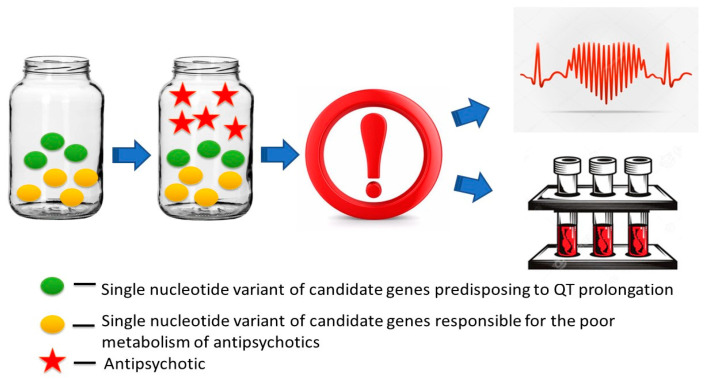
Genetic biomarkers of the risk of antipsychotic-induced long QT interval prolongation.

**Figure 12 ijms-23-15786-f012:**
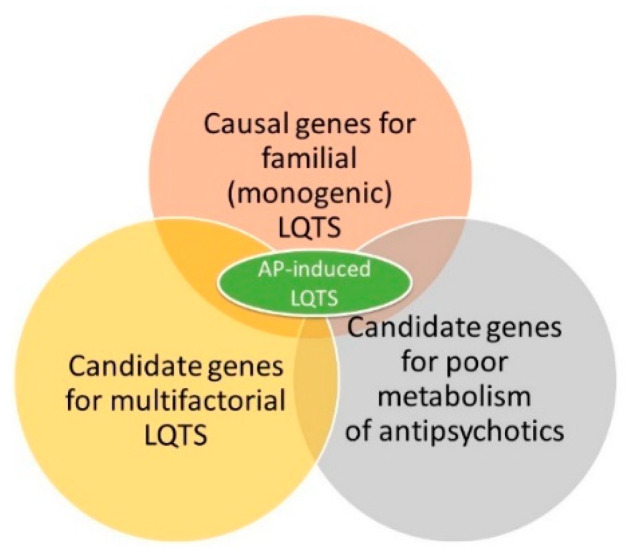
Evaluation of the cumulative risk of multifactorial long QT syndrome provoked or exacerbated by antipsychotics includes carriages of single nucleotide variants of genes predisposing to the development of long QT syndrome (LQTS) and carriages of single nucleotide variants of antipsychotic metabolism genes.

**Figure 13 ijms-23-15786-f013:**
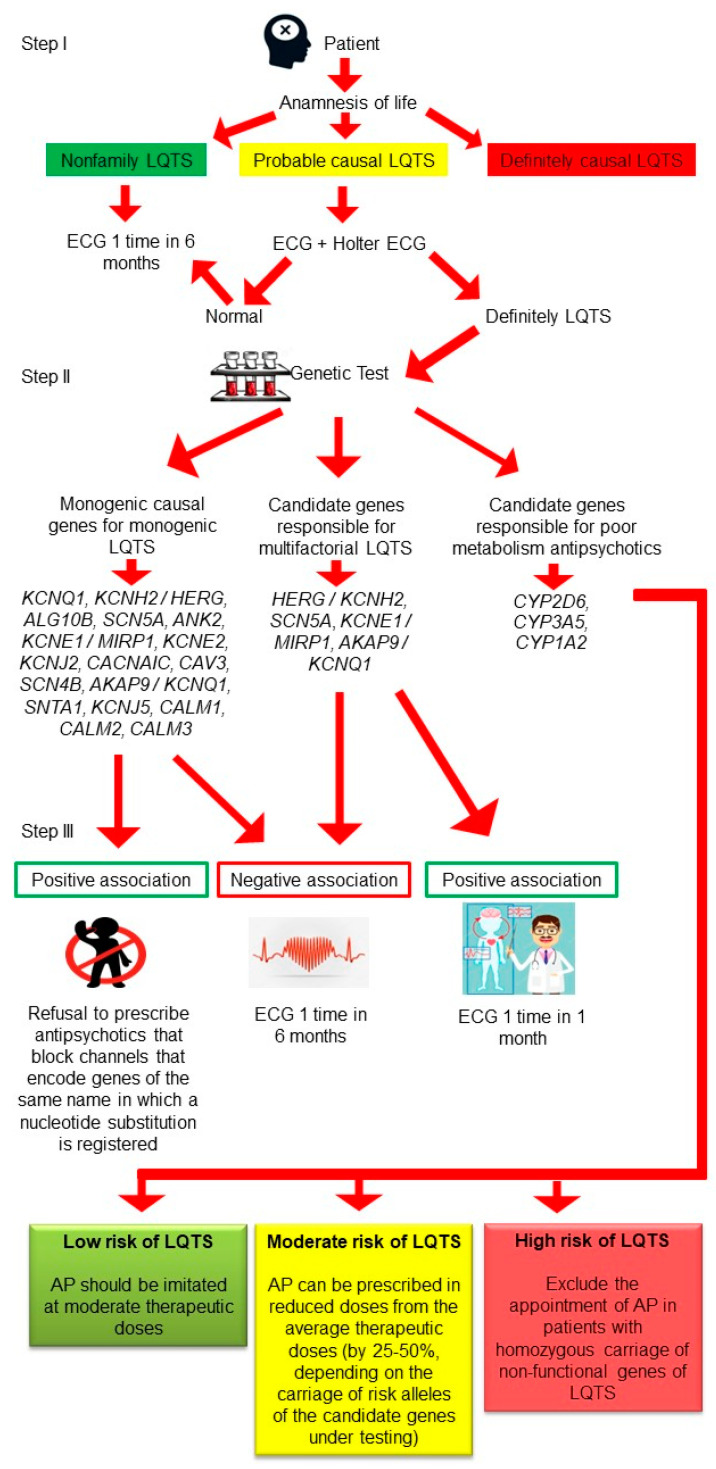
Algorithm for diagnosing the risk of developing long QT syndrome (LQTS) in schizophrenic patients requiring treatment with antipsychotics (APs). Note: ECG—electrocardiogram; IM—intermediate metabolizer; PM—poor metabolizer; UM—ultrarapid metabolizer.

**Table 1 ijms-23-15786-t001:** Casual genes for monogenic long QT syndrome.

Gene (OMIM Number)	Protein	Chromosome Location	Inheritance	Phenotype (OMIM Number)
*KCNQ1* (607542)	Potassium voltage-gated channel subfamily Q member 1	11p15.5-p15.4	AR	JLNS1 (220400)
			AD	LQT1 (192500)
*KCNH2/HERG* (152427)	Potential-dependent potassium channel H2 type	7q36.1	AD	LQT2 (613688)
*ALG10B* (603313)	Asparagine-associated glycosylation of 10 homologue B	12q22	AD	LQT2 (613688)
*SCN5A* (600163)	Alpha subunit of voltage-gated type 5 sodium channel	3p22	AD	LQT3 (603830)
*ANK2* (106410)	Ancrinin 2	4q25-q26	AD	LQT4 (600919)
*KCNE1/MIRP1* (176261)	Potential-dependent potassium channel E1 type	22q22.12	AR	LQT5 (613695) JLNS2 (612347)
*KCNE2* (603796)	Potential-dependent potassium channel E2 type	21q22.11	AD	LQT6 (613693)
*KCNJ2* (600681)	Voltage-gated potassium channel J2 type	17q24	AD	ATS or LQT7 (170390)
*CACNAIC* (114205)	Potential-dependent potassium channel alpha 1C subunit	12p13.33	-	BS or LQT8 (618447)
*CAV3* (601253)	Caveolin 3	3p25.3	AD	LQT9 (611818)
*SCN4B* (608256)	Voltage-dependent sodium channel beta 4 subunit	11q23.3	AD	LQT10 (611819)
*AKAP9/KCNQ1* (604001)	Anchor protein A-kinase 9 type	7q21.2	AD	LQT11 (611820)
*SNTA1* (601017)	Syntrophin alpha 1	20q11.21	AD	LQT12 (612955)
*KCNJ5* (600734)	Potential-dependent potassium channel J5 type	11q24.3	AD	LQT13 (613485)
*CALM1* (114180)	Calmodulin 1	14q32.11	AD	LQT14 (616247)
*CALM2* (114182)	Calmodulin 2	2p21	AD	LQT15 (616249)
*CALM3* (114183)	Calmodulin 3	19q13.32	AD	LQT16 (618782)

Note: AD—autosomal dominant; AR—autosomal recessive; BS—Brugada syndrome; LQT—long QT; ATS—Andersen-Tawil syndrome; JLNS—Jervell and Lange-Nielsen syndrome; OMIM—Online Mendelian Inheritance in Man [25].

**Table 2 ijms-23-15786-t002:** Mechanism of action and effect of antipsychotics used to treat schizophrenia on resting heart rate and conduction.

Antipsychotics	Mechanism of Action	Effect	References
High risk			
Chlorpromazine	Blocking I_Kr_ channels	QTc prolongation	[26,27,28]
Haloperidol			[29,30,31,32]
Ziprasidone			[32,33]
Moderate risk			
Amisulpride (oral)	Blocking I_Kr_ channels	QTc prolongation	[34,35]
Clozapine			[36,37]
Flupentixol			[38]
Haloperidol (oral)			[29,30,31,32]
Olanzapine			[26,39,40]
Pimozide			[32,40,41]
Quetiapine			[32,40,42]
Risperidone			[32,40]
Thioridazine			[28,32,40,43]
Mesoridazine			[32,33]
Perphenazine			[28]
Trifluoperazine			[28]
Sertindole			[40]
Sulpiride			[40]
Low risk			
Asenapine	Blocking I_Kr_ channels	QTc prolongation	[44]
Iloperidone			[44]
Paliperidone			[44]
Pimavanserin			[45]
Aripiprazole			[44]
No effect			
Aripiprazole	N/A	Do not cause QTc prolongation	[46]
Zuclopenthixol			[46]
Lurasidone			[46]

**Table 3 ijms-23-15786-t003:** Candidate genes and their single nucleotide variants predisposing to QT prolongation.

Gene (OMIM Number)	Protein	Chromosome Location	rsID	Effect	References
*KCNH2* (152427)	Potential-dependent potassium channel H2	7q36.1	rs189014161	Susceptibility to QT interval prolongation	[47]
*SCN5A* (600163)	Alpha subunit of voltage-gated type 5 sodium channel	3p22.2	rs1805124	Susceptibility to QT interval prolongation	[48]
*KCNE1/MIRP1* (176261)	Potential-dependent potassium channel E1 type	21q22.12	rs1805127	Susceptibility to QT interval prolongation	[49]
*AKAP9/KCNQ1* (604001)	Type 9 alpha kinase anchor protein	7q21.2	rs11772585	The T allele is associated with the risk of QT interval prolongation	[50]
			rs7808587	The GG genotype is associated with the risk of QT interval prolongation	[50]
			rs2282972	The T allele is associated with reducing the risk of QT interval prolongation	[50]
			rs2961024	The GG genotype is associated with the risk of QT interval prolongation	[50]

Note: OMIM—Online Mendelian Inheritance in Man [25].

**Table 4 ijms-23-15786-t004:** Candidate genes and their single nucleotide variants responsible for altering the pharmacokinetics of antipsychotics and leading to secondary antipsychotic-induced cardiac arrhythmias and conduction disorders.

Gene (OMIM Number)	Protein	Chromosome Location	rsID	Effect	References
*CYP2D6* (124030)	Isoenzyme 2D6 of cytochrome P450	22q13.2	rs16947, rs1135840, rs35742686, rs1135824, rs35742686, rs3892097, rs3892097, i4001456, rs3892097, i4001467 rs3892097, rs28371733, rs5030655, rs5030867, rs5030865, rs5030863, rs5030862, rs5030865, rs72549354	Risk of prolongation QT interval while taking chlorpromazine, haloperidol, perphenazine, thioridosine, aripiprazil, olanzapine, risperidone.	[72]
*CYP3A5* (605325)	Isoenzyme 3A5 of cytochrome P450	7q22.1	rs28365083, rs776746, rs10264272, rs41303343, rs55817950, rs28383479, rs41279854, rs72552791, rs56244447, rs28365085, rs28383468, rs41279857	Risk of prolongation QT interval while taking haloperidol, aripiprazole, risperidone.	[73]
*CYP1A2* (124060)	Isoenzyme 1A2 of cytochrome P450	15q24.1	rs2069514, rs12720461, rs2069526, rs56276455, rs72547516, rs28399424	Risk prolongation QT interval while taking olanzapine, clozapine, thioridazine, chlorpromazine.	[74]

Note: OMIM—Online Mendelian Inheritance in Man [25].

**Table 5 ijms-23-15786-t005:** Classification of psychotropic drugs according to the risk of QT prolongation interval and the development of arrhythmias [87,88].

Category	Characteristic
Class A	Drug without the risk of QT interval or TdP
Class B	Drug capable of QT interval prolongation
Class C	Drug with marked QT interval prolongation, reported cases of TdP or other serious arrhythmia

Note: TdP—fr. Torsades de pointes (pirouette-type tachycardia, polymorphic ventricular tachycardia in patients with a long QT interval).

## Data Availability

Not applicable.

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
