# Peer review of "Genetic Biomarkers of Antipsychotic-Induced Prolongation of the QT Interval in Patients with Schizophrenia"

_ijms, 2022, doi:10.3390/ijms232415786_

Round 1

Reviewer 1 Report

The review article from Vaiman et al. provided comprehensive information on the general concepts, genetic backgrounds, and physiological mechanisms underlying antipsychotic-induced LQT in schizophrenia patients. There are only a few very minor issues that may require some attention, as noticed below.

Table 3, header row: When using rsIDs, it is more common to say "SNP" than "SNV";

Figures 2 and 7 may need better resolutions, as the characters on the current version are not very clear.

Reviewer 2 Report

The authors did this narrative review on important topic of genetic biomarkers, associated on antipsychotic induced LQTS in patients with scizophrenia. Main part of this review focus on pharmacological predictors of LQTS  and presented data on causal genes for monogenic LQTS, candidate genes for multifactorial LQTS and candidate genes responsible for poor metabolism of antipsychotics.

Based on the summary of results, authors presented an algorithm for diagnosing the risk factors of developing LQTS in patients with schizophrenia with antipsychoics treatment (Table 13). 

My suggestion to include into abstract section more information about the results of this review and target reccomendations on evaluating risk of LQTS. 
